# An Improved In Vitro Blood-Brain Barrier Model for the Evaluation of Drug Permeability Using Transwell with Shear Stress

**DOI:** 10.3390/pharmaceutics16010048

**Published:** 2023-12-28

**Authors:** Junhyeong Kim, Seong-Ah Shin, Chang Sup Lee, Hye Jin Chung

**Affiliations:** 1College of Pharmacy and Research Institute of Pharmaceutical Sciences, Gyeongsang National University, Jinju 52828, Republic of Korea; jjun6914@gnu.ac.kr (J.K.); shinsaya@gnu.ac.kr (S.-A.S.); changsup@gnu.ac.kr (C.S.L.); 2Anti-Aging Bio Cell factory Regional Leading Research Center (ABC-RLRC), Gyeongsang National University, Jinju 52828, Republic of Korea

**Keywords:** blood-brain barrier, permeability, transwell, in vitro BBB model, shear stress, annular shaking dish, hCMEC/D3, immortalized human brain microvascular endothelial cell

## Abstract

The development of drugs targeting the central nervous system (CNS) is challenging because of the presence of the Blood-Brain barrier (BBB). Developing physiologically relevant in vitro BBB models for evaluating drug permeability and predicting the activity of drug candidates is crucial. The transwell model is one of the most widely used in vitro BBB models. However, this model has limitations in mimicking in vivo conditions, particularly in the absence of shear stress. This study aimed to overcome the limitations of the transwell model using immortalized human endothelial cells (hCMEC/D3) by developing a novel dish design for an orbital shaker, providing shear stress. During optimization, we assessed cell layer integrity using trans-endothelial electrical resistance measurements and the % diffusion of lucifer yellow. The efflux transporter activity and mRNA expression of junctional proteins (claudin-5, occludin, and VE-cadherin) in the newly optimized model were verified. Additionally, the permeability of 14 compounds was evaluated and compared with published in vivo data. The cell-layer integrity was substantially increased using the newly designed annular shaking-dish model. The results demonstrate that our model provided robust conditions for evaluating the permeability of CNS drug candidates, potentially improving the reliability of in vitro BBB models in drug development.

## 1. Introduction

Drug development is a time-consuming and expensive process. The development of drugs that target the central nervous system (CNS) is particularly challenging. The development period of CNS drugs has been, on average, 20% longer and the approval time has been 38% longer than that of other approved drugs (from 2000 to 2017) in the United States [1]. CNS-targeted drug development is more demanding than the development of other drugs for several reasons, with one of the predominant challenges being crossing the Blood-Brain barrier (BBB).

The BBB is a selectively permeable membrane composed of endothelial cells, astrocytes, pericytes, and microglial cells. It serves as a physical and enzymatic defense system that isolates the brain from the rest of the body, selectively permitting certain molecules to pass through while restricting exposure to toxic substances. This is a significant hurdle that prevents the diffusion of several chemicals into the brain [2]. For CNS drug candidates to be successful, it is imperative to ensure that they are well delivered to the brain across the BBB and sufficiently exposed to the target site. Furthermore, predicting brain penetration during the developmental process is important for identifying the most effective candidates among numerous lead compounds. It is noteworthy that results from commonly used animal models in preclinical studies often exhibit a low correlation with human responses. This limitation makes predicting the actual effects of drugs on humans challenging [3,4].

Therefore, continuous research efforts directed toward the development of highly predictive models using human cells aim to bridge the gap between in vitro investigations and in vivo applications [5,6]. Numerous in vitro BBB models have been developed, each with its advantages and limitations. These include the two-dimensional transwell model and three-dimensional static models based on hydrogels, spheroids, and organoids [7,8]. Additionally, there are 3D models with the fluid flow, using a microfluidic pump [9] or bidirectional flow [10]. Among them, the widely studied microfluidic models allow for the application of shear stress, nondestructive microscopy, and the flexibility of parallelization that mimics the physiological organ architecture [11]. However, the requirement of specialized equipment for shear stress applications and the intricate inclusion of electrodes into the platform in order to assess transendothelial electrical resistance (TEER) is challenging. Consequently, these models may not be suitable for linear kinetic experiments and face challenges when screening a large number of drug candidates or formulations to assess BBB permeability [12]. However, the commonly used transwell models offer simplicity in terms of fabrication and ease of use, making them well suited for high-throughput screening (HTS) owing to their high processing capacity. Transwell models have notable advantages, including easy TEER measurement and straightforward applications in various culture methods, such as monocultures and multiple cultures [13,14]. Nevertheless, it has its own constraints that primarily originate from the difficulty in applying shear stress. Shear stress is crucial for developmental and physiological processes in the formation of endothelial cell vasculature and the regulation of the expression of various proteins [15,16].

We developed a novel in vitro BBB model suitable for permeability assessment by applying shear stress to an HTS-friendly transwell system using an orbital shaker. Through the design of a novel annular shaking dish (AS), we achieved a model with greater cell-layer integrity, using immortalized cells, with consistent experimental results.

## 2. Materials and Methods

### 2.1. Chemicals and Reagents

Ascorbic acid, atenolol, dantrolene sodium, gabapentin, hydrocortisone, Ko143, metoprolol tartrate, naproxen, phenytoin sodium, sulpiride, propranolol HCl, type I rat tail collagen solution, lucifer yellow CH dipotassium salt (LY), fluorescein isothiocyanate (FITC)–dextran (4 and 40 kDa), rhodamine 123, and 7.5% sodium bicarbonate solution were purchased from Sigma-Aldrich (St. Louis, MO, USA). Midazolam was purchased from Cerilliant (Austin, TX, USA). Fexofenadine hydrochloride, carbamazepine, donepezil HCl, and cyclosporine A were purchased from Tokyo Chemical Industry (Tokyo, Japan).

Molecular, Cellular, and Development Biology 131 medium (MCDB131 medium), fetal bovine serum (FBS), basic fibroblast growth factor (bFGF), chemically defined lipid concentrate, N-2-hydroxyethylpiperazine–N′-2-ethanesulfonic acid (HEPES), Hanks’ balanced salt solution (HBSS), 0.25% trypsin-EDTA, penicillin-streptomycin, and Insulin-transferrin-selenium-ethanolamine (ITS) were purchased from Gibco (Grand Island, NY, USA). HPLC grade methanol and water were purchased from Fisher Scientific Korea (Seoul, Republic of Korea).

### 2.2. Cell Culture

hCMEC/D3, the human cerebral microvascular endothelial cell line, was obtained from Cellution Biosystems Inc., (Burlington, ON, Canada), and U87MG was obtained from HTB-14™, ATCC (Manassas, VA, USA). The cells were cultured in MCDB131 medium with the following components: 5% (*v*/*v*) FBS, 1 ng/mL bFGF, 1% (*v*/*v*) chemically defined lipid concentrate, 10 mM HEPES, 1.4 µM hydrocortisone, 5 µg/mL ascorbic acid, and 1% (*v*/*v*) penicillin–streptomycin (the final concentration was 100 U/mL of penicillin and 100 μg/mL of streptomycin).

A T-75 flask for hCMEC/D3 was coated with type I rat tail collagen at a concentration of 150 µg/mL in distilled water for one hour in the incubator. After aspirating the collagen solution, the T-75 flask was rinsed with warm phosphate buffered saline (PBS) for seeding. The seeding densities were 25,000 cells/cm^2^ for hCMEC/D3 and 15,000 cells/cm^2^ for U87MG, respectively. The hCMEC/D3 cells and U87MG cells were incubated in a humidified 5% CO_2_ environment at 37 °C. The medium for each cell line was replaced two to three times per week.

### 2.3. Culture for Transwell Inserts

The culture inserts (Costar^®^, Washington, DC, USA, 6.5 mm diameter) of polycarbonate (PC) and polyester (PET) membrane were procured from Corning Inc. (Corning, NY, USA). Culture inserts were coated with 50 µL of type I rat tail collagen at a concentration of 180 µg/mL in 70% ethanol and dried overnight under UV light. After washing twice with PBS, the cells were seeded at a density of 1 × 10^5^ hCMEC/D3 cells/well in the apical chamber and 3 × 10^4^ U87MG cells/well on the basolateral side of the insert. The cultivation method was modified from that described in the literature [17]. Briefly, the insert was inverted, and U87MG cells were initially seeded. The insert was flipped 3–4 h after seeding the U87MG cells. Subsequently, hCMEC/D3 cells were seeded 1 day later.

### 2.4. Optimization of the Transwell Model

We investigated the cell layer integrity to optimize various conditions of the transwell BBB model. The method was modified from an existing protocol [18]. We present this briefly, as follows:

#### 2.4.1. Trans-Endothelial Electrical Resistance (TEER)

The TEER values were measured using an STX2 chopstick electrode with EVOM2™ (World Precision Instruments, Sarasota, FL, USA), following the manufacturer’s instructions, to evaluate cell layer integrity. The measurements obtained were corrected by subtracting the empty insert value and multiplying it by the surface area of the culture insert (0.33 cm^2^) to derive the final TEER values.

#### 2.4.2. Lucifer Yellow (LY) Diffusion

To evaluate the cell layer integrity, the permeability of LY, a passive paracellular transport marker, was measured. LY was used at a concentration of 200 μM in transport buffer (HBSS with 10 mM HEPES, 350 mg/L sodium bicarbonate, and 25 mM glucose). Before each experiment, the cells were washed twice with a pre-warmed transport buffer. After a 30 min incubation, LY was added to the apical side (AP) at a concentration of 200 μM. The LY permeated to the basolateral side (BL) was measured after a 1 h incubation at 37 °C with shaking at 100 rpm. Each sample was measured at wavelengths of 480 nm excitation and 530 nm emission with the VICTOR^®^ Nivo multimode plate reader (PerkinElmer Inc., Sarasota, FL, USA). The %LY diffusion value was calculated using Equation (1):% LY diffusion = 100 × ((LY_BL_ × V_BL_)/(LY_AP_ × V_AP_))(1)
where:LY_BL_ is the concentration of LY in the basolateral receiver chamber (μM)V_AP_ is the volume in the apical receiver chamber (cm^3^)LY_AP_ is the concentration of LY added to the apical donor chamber (μM)V_BL_ is the volume in the basolateral donor chamber (cm^3^)

#### 2.4.3. Rhodamine 123 Efflux

Rhodamine 123, known as a P-glycoprotein (P-gp) substrate, was used at a concentration of 10 μM in transport buffer to assess the P-gp activity. Bidirectional permeability tests were performed at time points of 0, 30, 60, and 90 min with shaking at 100 rpm. Each sample was measured at 480 nm/530 nm (excitation/emission wavelengths) using the VICTOR^®^ Nivo multimode plate reader. The apparent permeability coefficient (P_app_) was determined using Equation (2):P_app_ (cm/s) = dQ/dt × 1/(A × C_0_)(2)
where:A is the surface area of the insert filter membrane (cm^2^)C_0_ is the initial concentration in the donor solution (μM)dQ/dt is the amount of the drug transported within a given time period (pmol/s)

The efflux ratio of rhodamine123, calculated from the obtained P_app_ values using Equation (3), was used to evaluate the P-gp activity.
Efflux ratio = P_app, BL to AP_/P_app, AP to BL_(3)

### 2.5. Quantitative Real-Time PCR

After a 17-day incubation, the culture inserts were rinsed with PBS and the cells were harvested. Total RNA was extracted using the TRIzol^®^ reagent (Invitrogen, Waltham, MA, USA), followed by complementary DNA synthesis performed from the isolated total RNA using a SimpliAmp Thermal Cycler (Applied Biosystems Co., Waltham, MA, USA). Subsequently, quantitative real-time polymerase chain reaction (real-time PCR) was conducted on a Step One Plus Real-time PCR System Cycler (Applied Biosystems Co., Waltham, MA, USA) using a Power SYBR Green PCR mix (Applied Biosystems Co., Waltham, MA, USA). GAPDH served as the reference gene for the PCR primer sequences [19,20,21], which are detailed in Appendix A.

### 2.6. Efflux Transporter (P-gp, BCRP) Inhibition Assay

A modified method from a previously published paper [19] was used for the inhibition assay. Briefly, rhodamine 123, a P-gp substrate, and dantrolene, a breast cancer resistance protein (BCRP) substrate, were used at a concentration of 10 μM. Cyclosporin A (10 μM), a P-gp inhibitor, and Ko143 (2 μM), a BCRP inhibitor, were pre-incubated for 30 min in both chambers. The bidirectional permeability of substrates was evaluated by measuring changes in P_app_ and the efflux ratio for both the inhibitor- and vehicle-treated groups. The concentration of dantrolene was determined using liquid chromatography with tandem mass spectrometry (LC-MS/MS).

### 2.7. Permeability Test

For the permeability assessment, FITC-dextran was utilized at a concentration of 1 mg/mL, while rhodamine 123 was used at a concentration of 10 μM. All other drugs, excluding FITC-dextran and rhodamine 123, were prepared at a concentration of 20 μM in the transport buffer. To prevent nonspecific binding, all compounds were pre-incubated for 30 min prior to the permeability study. The samples were collected at 0 min from the donor and at 30, 60, and 90 min from the acceptor and immediately replaced with a new transport buffer. Incubation was carried out in a humidified 5% CO_2_ environment at 37 °C with shaking at 100 rpm. Concentrations of the compounds were determined using LC-MS/MS, with the exception of FITC-dextran and rhodamine 123. The FITC-dextran and rhodamine 123 concentrations in the samples were measured by a VICTOR^®^ Nivo multimode plate reader.

### 2.8. Annular Shaking Dish (AS)

Drawing inspiration from the published literature [22,23] on the utilization of an orbital shaker, novel ASs were designed to apply an optimal fluid flow to culture inserts. A 15 cm AS was created by attaching a 10 cm culture dish inverted onto a 15 cm culture dish, followed by the attachment of an acrylic plate on top (Figure 1a). Using the same method, a 10 cm AS was constructed by stacking a 6 cm dish onto a 10 cm dish. When each component was attached, polydimethylsiloxane was used in a ratio of 10:2. The assembled dish was sterilized with 70% ethanol, followed by UV irradiation for 24 h.

After culturing the cells in culture inserts, as described in Section 2.3, they were transferred to the AS prepared on day 3, and cultivation was carried out with orbital shaking at 150 rpm until day 17 (Figure 1b). The cultivated AS BBB model was employed to investigate the effects of the AS size, as well as the influence of FBS and ITS.

### 2.9. LC-MS/MS

Compound concentrations were determined using an Agilent 1260 HPLC system coupled to an Agilent 6460 triple-quadrupole mass spectrometer (Agilent Technologies, Singapore) and a Shimadzu UFLC XR instrument (Shimadzu, Tokyo, Japan) coupled to a TSQ Quantum Ultra triple quadrupole mass spectrometer (Thermo Fisher Scientific, Bremen, Germany). Chromatographic separations were performed on Agilent Poroshell 120 EC-C18 (3.0 × 50 mm, 2.7 µm; Agilent Technologies, Inc., Santa Clara, CA, USA) and YMC triart C18 (2.0 × 50 mm, 3 µm; YMC Co., Ltd., Kyoto, Japan) columns using 5 mM ammonium formate in water (pH 4) and methanol. The detailed analysis conditions are listed in Appendix A.

### 2.10. Statistics

Changes in TEER values over the cultivation period were statistically analyzed using repeated-measures one-way ANOVA. Unpaired *t*-tests were used for the comparison of means between two groups, whereas one-way ANOVA was used for the comparison of means between three or more groups. Statistical analysis was conducted with GraphPad Prism 5.

## 3. Results

### 3.1. Optimization of the 24-Transwell Model

To optimize various factors, including culture methods and culture inserts, affecting the 24-transwell BBB model, factors were assessed based on TEER values and %LY diffusion. Initially, the cultivation method was optimized, and mono- and co-cultures were compared using the culture inserts of the PC membrane. 

The TEER values for the hCMEC/D3 mono-culture group and the co-culture group of hCMEC/D3 with U87MG were 8.0 ± 0.5 and 9.9 ± 0.6 Ω·cm^2^, respectively, showing statistical significance (Figure 2a). The %LY diffusion was 11.4 ± 0.4% and 8.4 ± 0.8% for the mono-culture and co-culture, respectively, indicating a higher cell layer integrity in the co-culture group (Figure 2b).

To apply fluid flow, the mono-cultured inserts were incubated in a 24-well plate on an orbital shaker. The use of orbital shaking also resulted in an increase in the TEER value to 7.0 ± 1.0 Ω·cm^2^ at day 11 compared to that (4.6 ± 0.3 Ω·cm^2^) of the no shaking group (Figure 2c). Additionally, the shaking group showed a significant decrease in the leakage of LY at 10.1 ± 0.3% compared to that (14.2 ± 1.2%) of the no shaking group (Figure 2d).

To assess the effect of culture inserts, experiments were conducted by altering both the membrane material and pore size, employing the co-culture method with orbital shaking. The membrane type, whether PET or PC, did not exhibit any significant differences in TEER and %LY diffusionTherefore, for further experiments, we have used PC membranes. However, the 3 μm pore size group exhibited higher cell layer integrity on day 11, with a TEER value of 21.5 ± 1.2 Ω·cm^2^ and %LY diffusion value of 3.7 ± 0.3%, compared to the 0.4 μm pore size group, with a TEER value of 9.0 ± 0.2 Ω·cm^2^ and %LY diffusion value of 7.0 ± 0.6% (Figure 2e,f).

Although there was a significant improvement in the 24-well model, it fell short of the commonly accepted threshold of 3% for %LY diffusion, indicating the robustness of a typical monolayer [18]. The absence or insufficiency of shear stress may have led to a lack of cell layer integrity.

### 3.2. Annular Shaking Dish (AS)

#### 3.2.1. AS Effect on Cell Layer Integrity

Building on prior research [22,23], a novel AS was designed to optimally apply fluid flow to the culture insert. The modulation of shear stress can be achieved by adjusting parameters such as the dish size, radius of orbital rotation, rotational speed (expressed in RPM) of the shaker, and viscosity. In humans, shear stress has been reported to range from 4 to 30 dyn/cm^2^ for arterial circulation and from 1 to 4 dyn/cm^2^ for venous circulation [24]. In addition, 10 dyn/cm^2^ was the best shear stress for a high junctional protein intensity, as reported previously [25]. The rotational speed of the orbital shaker was selected by calculating the maximum shear stress using Equation (4):(4)$$\tau_{max}=a\sqrt{\rho \eta {\left(2\pi f\right)}^{3}}$$
with
a: the radius of orbital rotation (1.9 cm)*ρ*: the density of the medium (g/mL)*η*: the viscosity of the medium (7.78 × 10^−3^ dyn·s/cm^2^)*f*: the frequency of rotation (rotations/s)

The calculated result indicated that a value of 10 dyn/cm^2^ was achieved at 150 rpm using a 1.9 cm radius of the orbital shaker.

We examined the effects of the dish size (24 well plate, 10 cm and 15 cm AS) while keeping the rotational speed (150 rpm) and shaking radius (1.9 cm) constant. For cell cultivation, optimized conditions (3 μm pore size, PC membrane, co-cultured with U87, with shaking) were employed. TEER measurements were conducted after transferring all culture inserts to 24-well plates. As a result of the increasing shear stress associated with the dish size, an elevation in the TEER value was observed with the highest value of 51.5 ± 0.9 Ω·cm^2^ (Figure 3a), concomitant with a reduction in the %LY diffusion at 1.7 ± 0.1% (Figure 3b). Interestingly, we also observed a significant increase in P-gp activity in the 15 cm AS group, confirmed by the efflux ratio (Figure 3c) exceeding two for rhodamine 123. Using the newly designed dish with orbital shaking, we confirmed the alleviation of leaky membranes of hCMEC/D3.

#### 3.2.2. ITS and FBS Effects on the Cell Layer Integrity

Based on previous literature suggesting that a low FBS concentration promotes differentiation [26,27], we aimed to adjust the FBS concentration using the 15 cm AS. Before this study, we investigated the effects of ITS, an additive that supports cell survival and division. It is commonly used to reduce the FBS concentration. We found that ITS positively affected cell layer integrity (Figure 4). On day 17, the TEER value was 79.8 ± 5.3 Ω·cm^2^ in the 5% FBS + ITS-treated group and 51.4 ± 1.5 Ω·cm^2^ in the 5% FBS group (Figure 4a). While the %LY leakage values were below 3% and not statistically different in either the 5% FBS and 5% FBS + ITS groups, the ITS-treated group exhibited a lower %LY diffusion at 0.5 ± 0.1% compared to the ITS-non-treated group (1.2% ± 0.2%) (Figure 4b). Furthermore, a significant increase in P-gp activity was observed in 5% FBS + ITS group at a value of 4.0 ± 0.04, compared to the 5% FBS group at 2.4 ± 0.05 (Figure 4c).

The 0% FBS + ITS treatment group failed to form a robust cell layer, resulting in a low TEER and high %LY leakage. A negligible difference was observed between TEER and% LY diffusion values in the 2% and 5% FBS + ITS groups. However, a significant difference was observed in the efflux ratio of rhodamine 123 (Figure 4c). The efflux ratio in the 2% FBS + ITS group was higher at 6.73 ± 0.94, compared to 4.04 ± 0.04 in the 5% FBS + ITS group (*p* < 0.001). This result aligns with the existing literature [26,27], demonstrating increased differentiation when the concentration of FBS is reduced.

### 3.3. The Evaluation of the Optimized Annular Shaking Dish (AS) BBB Model

The final optimized conditions were as follows: co-culture, culture insert with a 3 μm pore size and polycarbonate membrane, 2% FBS + ITS, and 15 cm cell culture dish with orbital shaking at 150 rpm. The optimized AS BBB model was evaluated as follows.

#### 3.3.1. mRNA Levels of the Junctional Proteins

To investigate whether the experimentally observed increase in TEER values and decrease in %LY permeability were associated with the expression of tight junction proteins, quantitative analysis of mRNA levels was performed using quantitative real-time PCR. Real-time PCR analysis included the mRNA of the tight junction proteins (claudin-5 and occludin) and adhesive junction protein (vascular endothelial (VE)-cadherin).

Upon normalization to GAPDH, the mRNA levels of claudin-5 and occludin in the AS model were approximately 300-fold and 40-fold higher, respectively, than those in the static monoculture model (Figure 5). This result is consistent with the enhanced cell-layer integrity confirmed experimentally. This is also consistent with prior research suggesting a crucial role for claudin-5 and occludin in BBB permeability [28]. While the mRNA level increased, there was no statistically significant difference in the mRNA levels of VE-cadherin.

#### 3.3.2. Inhibition Assay of P-gp and BCRP

To assess the efflux transporter activity in the AS BBB model, an inhibition assay was conducted. Rhodamine 123 served as the substrate for P-gp, whereas dantrolene was used as the substrate for BCRP. Treatment with the P-gp inhibitor, cyclosporin A, resulted in a notable reduction in the efflux ratio of the P-gp substrate from 4.2 ± 0.15 to 1.6 ± 0.13 (Figure 6). Moreover, the treatment with the BCRP inhibitor, Ko143, also manifested a significant reduction in the efflux ratio of the BCRP substrate from 2.25 ± 0.69 to 0.33 ± 0.03. Thus, we confirmed that the activity of the efflux transporters necessary for the assessment of BBB permeability was sufficient.

#### 3.3.3. Permeability Assay

Using the optimized AS model, we tested the permeability of 14 substances (Table 1). The measured permeability was represented in terms of the P_app_ values calculated using Equation (2). The permeability of P-gp substrates such as fexofenadine and rhodamine 123 was low, and dextran molecules with molecular weights of 4 and 40 kDa showed minimal penetration (Figure 7a). In addition, drugs known for their effective transport into the brain, such as donepezil and carbamazepine, exhibited high permeability.

To compare the measured in vitro results with the in vivo data, we collected unbound plasma-cerebrospinal fluid (CSF) partition coefficient (K_p,uu,CSF_) values from published studies [29,30,31,32]. The K_p,uu,CSF_ values are usually defined as the ratio of the free CSF concentration to the free plasma concentration at equilibrium [33]. These values are presented in Table 1 and Figure 7. Although the number of drugs used for the comparison was limited, preventing a precise confirmation of linearity, there was a noticeable trend indicating that drugs with higher P_app_ values tended to exhibit high in vivo K_p,uu,CSF_ values.

## 4. Discussion

When developing CNS drugs, it is crucial to evaluate the extent to which candidate compounds permeate the BBB. Therefore, the development of dedicated tools for this purpose is imperative. The use of in vitro BBB models based on human cells has the potential to effectively screen promising candidates.

In this study, we created a transwell-based BBB model suitable for HTS using immortalized human cells. Immortalized human cells offer the advantages of low variability, fewer ethical considerations, and ease of use. The immortalized cells used were the human cerebral microvascular endothelial cell line, hCMEC/D3, derived through the co-expression of human telomerase reverse transcriptase and Simian Vacuolating Virus 40 large T antigen. The hCMEC/D3 cells are well known for drug uptake and active transport in in vitro BBB models [34]. However, the absence of shear stress is a major limitation of the transwell models using immortalized cells because shear stress is crucial for differentiation and various physiological processes in vascular formation [15,35]. Overcoming this limitation could lead to significant improvements in various culture insert models that utilize immortalized, primary, and induced pluripotent stem cells (iPSC).

When shear stress was applied to the 24-well transwell using an orbital shaker, we observed a subtle yet significant difference in cell layer integrity (Figure 2c,d). The fluid flow generated by the orbital shaker differs between the center and periphery of the well [36]. In the central region, significantly reduced shear stress occurred, and the observed differences between the shaking and no-shaking groups could be attributed to the shear stress near the wall of the culture insert. 

We devised a dish that allowed the model to apply shear stress more effectively using an orbital shaker. Using the AS, we enhanced the cell layer integrity, achieving a remarkable maximum TEER value of 78.9 ± 9.7 Ω·cm^2^ (Figure 4a). This TEER value significantly exceeds the values reported in previous publications [37,38,39], showing a substantial improvement. The published models in the existing literature involved static conditions with a low integrity of the cell layer, utilizing the immortalized hCMEC/D3 cell line in a transwell model [37,38]. In these models, TEER values were below 15 Ω·cm^2^, indicating a leaky cell layer. To overcome this, a new model was developed by applying the dynamic micro tissue engineering system [40] to introduce fluid flow to the culture insert [39]. However, even in this model, TEER values did not exceed 20 Ω·cm^2^, and a low integrity of the cell layer was observed. We developed an AS capable of applying shear stress to culture inserts using an orbital shaker without any specific machine. Through this innovation, we achieved a high cell layer integrity. These results are in concordance with the mRNA expression of junctional proteins, especially claudin-5 and occludin.

Claudins play a crucial role in maintaining the barrier function, and claudin-5, especially, is the most abundant tight junction protein in the BBB. Occludin contributes to the integrity of tight junctions and is strongly associated with the permeability of substances across the BBB [28]. VE-cadherin, an endothelium-specific adhesion molecule, is crucial for controlling vascular permeability and maintaining endothelial cell contact [41]. When comparing the mRNA expression levels between the static monoculture models and our optimized AS model, the mRNA levels of claudin-5 and occludin significantly increased, while VE-cadherin also showed an increase, although not statistically significant (Figure 5). The increase in mRNA aligns with enhanced cell layer integrity, as confirmed by increased TEER values and decreased %LY diffusion. This implies that increased integrity may stem from the upregulation of tight junction proteins, including claudin-5 and occludin. This is particularly significant, as it represents an important achievement for a BBB permeability model using immortalized endothelial cells.

Furthermore, the activities of P-gp and BCRP were confirmed by assessing changes in the permeability of substrates in the presence and absence of inhibitors. P-gp and BCRP are extensively present in the human BBB and significantly limit the transport of various substances to the brain [42,43,44]. A sufficient level of activity was observed in the inhibition assay, as evidenced by reduced efflux ratios in the presence of inhibitors.

Using the permeability assay, the in vitro AS BBB model could distinguish between compounds with low and high permeabilities (Table 1). The P_app_ values exhibited a greater than 180-fold difference between drugs that effectively penetrated the brain and those that did not, confirming the model’s ability to distinguish between compounds with low and high brain permeability (Figure 7a). This confirmed that this model is suitable for use in permeability tests. Furthermore, although not entirely appropriate for comparisons because of species differences, the tendency was found by comparisons between the in vitro P_app_ results and in vivo K_p,uu,CSF_ values (Figure 7b).

The limitation of our study lies in the limited number of drug permeability evaluations required for establishing comparisons with in vivo data, making it challenging to predict human permeability values. Additionally, because the in vivo data used were K_p,uu,CSF_ from animals, not represented by K_p,uu,Brain_, or Q_ECF_ in the human brain [19], using this study as a tool for predicting real values requires the incorporation of modeling or simulations. Furthermore, the viscosity in the medium we used was relatively low compared to that of human blood (0.035–0.055 dyn·s/cm^2^), making it a significant factor influencing shear stress [45]. This could potentially affect the distribution of the actual drug [46,47]. Moreover, the TEER values were still lower compared to those observed in vivo and the results (>1500 Ω·cm^2^) obtained using induced pluripotent stem cells (iPSCs) [48,49], although the transport of LY through the paracellular pathway was remarkably low, measuring less than 1%. Despite these constraints, our study offers valuable insights into shear stress in various BBB models using the transwell model. Additionally, the AS can be loaded onto an orbital shaker in multiples, with each AS containing several culture inserts simultaneously. These features make it potentially suitable for HTS.

In conclusion, we established an in vitro BBB model for a permeability assay using hCMEC/D3 cells with a novel AS. We showed that the cell layer integrity was significantly increased using AS with orbital shaking. Based on the results of the junctional protein mRNA analysis, sufficient integrity was achieved for the permeability test. Using this permeability assay, the in vitro AS BBB model distinguished between low- and high-brain-permeability compounds. Therefore, this AS-BBB model may be an effective tool in the HTS of CNS-drug candidates. 

## Figures and Tables

**Figure 1 pharmaceutics-16-00048-f001:**
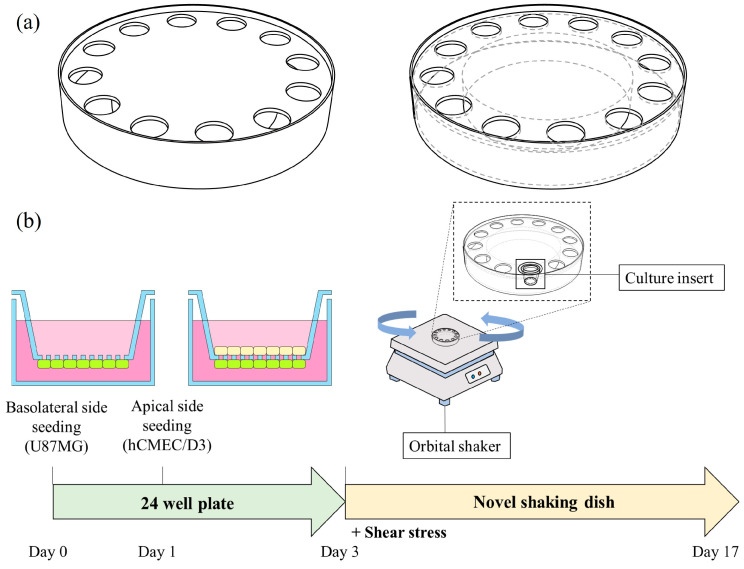
A detailed schematic illustration of (**a**) the annular shaking dish (AS) and (**b**) the culture protocol for the AS BBB model.

**Figure 2 pharmaceutics-16-00048-f002:**
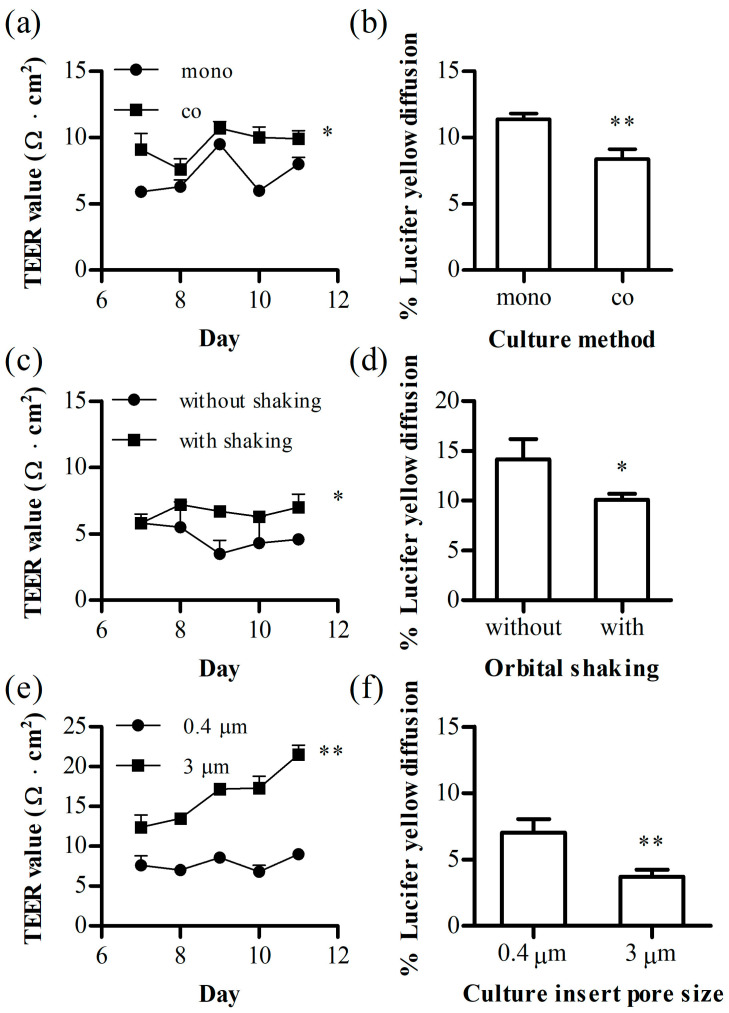
The optimization of the BBB 24-transwell model using the trans-endothelial electrical resistance (TEER) value and %lucifer yellow (LY) diffusion. (**a**,**b**) Effect of the co-culture method on the TEER value and %LY diffusion. (**c**,**d**) Influence of orbital shaking at 150 rpm on the TEER value and %LY diffusion. (**e**,**f**) Effect of the pore size of the culture inserts on the TEER value and %LY diffusion. Each value represents the mean ± standard deviation (*n* = 3). *, *p* < 0.05, **, *p* < 0.01.

**Figure 3 pharmaceutics-16-00048-f003:**
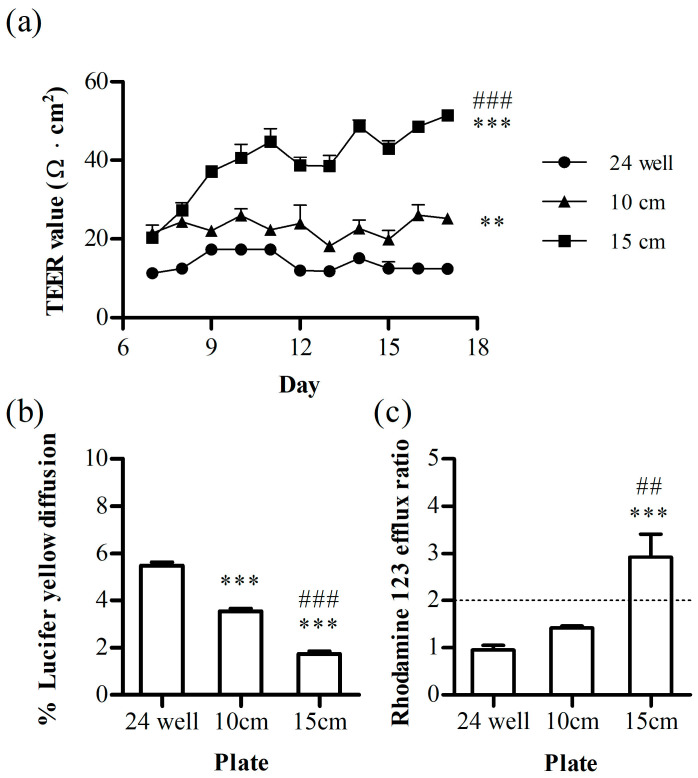
The annular shaking dish (AS) size effect on the (**a**) trans-endothelial electrical resistance (TEER), (**b**) %lucifer yellow (LY) diffusion, and (**c**) rhodamine 123 (P-gp substrate) efflux ratio. Each bar represents the standard deviation (*n* = 3). **, *p* < 0.01, ***, *p* < 0.001 vs. 24 well, ##, *p* < 0.01, ###, *p* < 0.001 vs. 10 cm.

**Figure 4 pharmaceutics-16-00048-f004:**
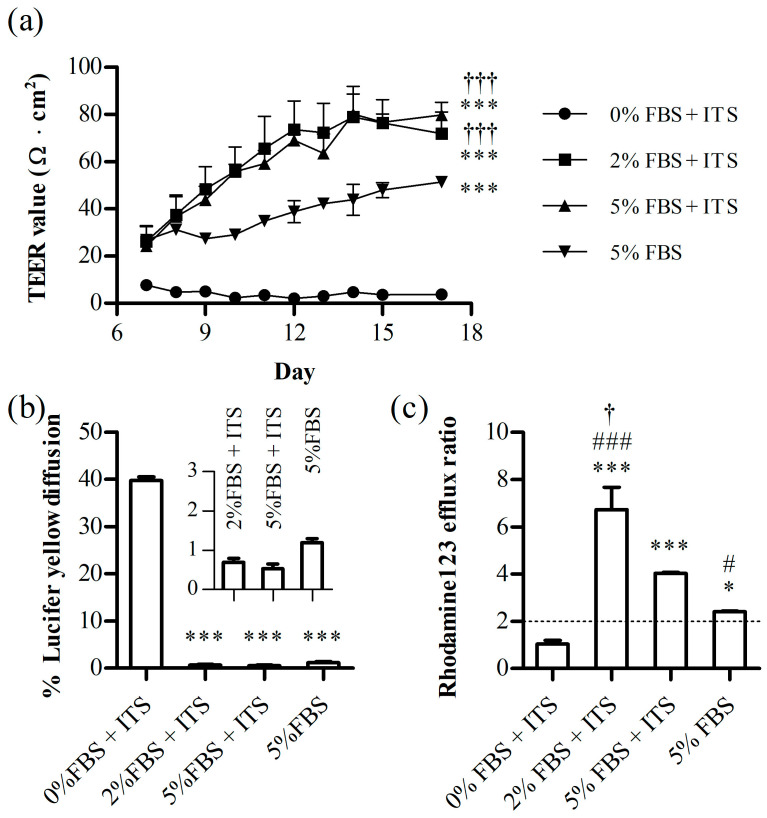
The FBS and ITS effect on the (**a**) trans-endothelial electrical resistance (TEER) value, (**b**) %lucifer yellow (LY) diffusion, and (**c**) rhodamine123 (P-gp substrate) efflux ratio. Each bar represents the standard deviation (*n* = 3). *, *p* < 0.05, ***, *p* < 0.001 vs. 0% FBS, #, *p* < 0.05, ###, *p* < 0.001 vs. 5% FBS + ITS, †, *p* < 0.05, †††, *p* < 0.001 vs. 5% FBS.

**Figure 5 pharmaceutics-16-00048-f005:**
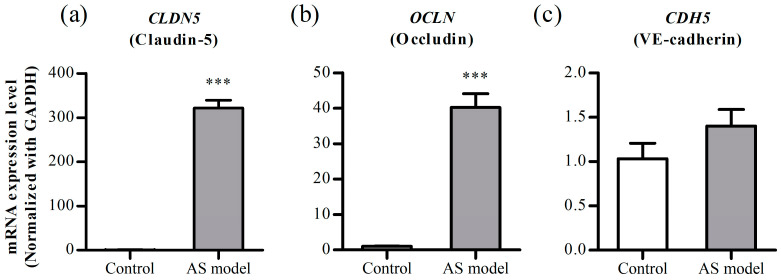
The mRNA expression levels of *CLDN5* (**a**), *OCLN* (**b**), and *CDH5* (**c**) with the annular shaking dish (AS) model and mono-static model (Control) using quantitative real-time PCR. Each value is the mean value normalized with GAPDH. Bars represent the standard deviation (*n* = 3). ***, *p* < 0.001.

**Figure 6 pharmaceutics-16-00048-f006:**
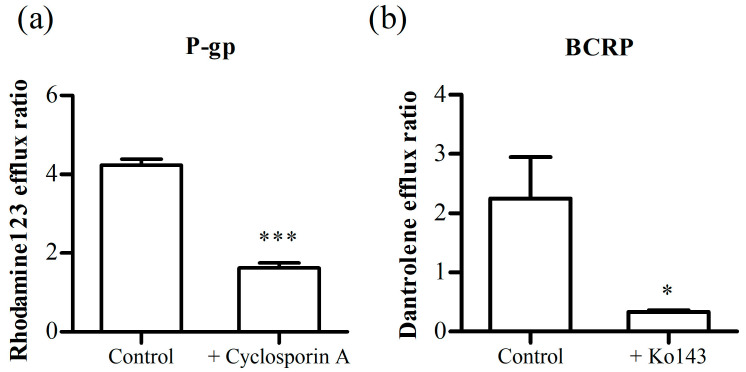
Efflux ratios for (**a**) rhodamine 123 and (**b**) dantrolene were determined with and without inhibitors, where 10 μM cyclosporin and 1 μM Ko143 were employed as P-gp and BCRP inhibitors, respectively. Bars represent the standard deviation (*n* = 3). *, *p* < 0.05, ***, *p* < 0.001.

**Figure 7 pharmaceutics-16-00048-f007:**
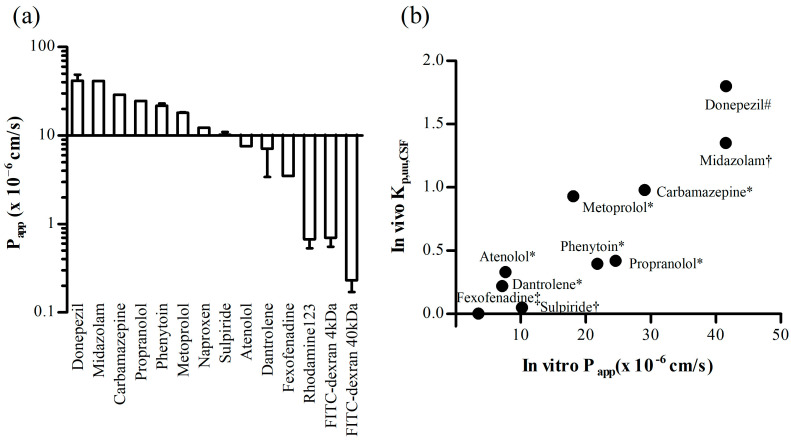
(**a**) The mean measured Blood-Brain barrier (BBB) permeability (P_app, AP to BL_) of various drugs using an in vitro AS BBB model. The used concentrations were 10 μM for rhodamine 123, 1 mg/mL for FITC-dextran, and 20 μM for the other drugs. Bars represent the standard deviation (*n* = 3). (**b**) The comparison between the in vitro P_app_ value and in vivo K_p,uu,CSF_ values [29,30,31,32] from different species including rat, mouse, monkey, and human. Each value represents the mean value. The in vivo K_p,uu,CSF_ values for the following symbols were obtained from other studies: * [31], # [29], † [30], ‡ [32].

**Table 1 pharmaceutics-16-00048-t001:** The P_app_ values and in vivo K_p,uu,CSF_ of the compounds used. The measured P_app_ value represents the mean ± standard deviation (*n* = 3). The P_app_ values were measured from an in vitro AS BBB model, and in vivo K_p,uu,CSF_ values were obtained from the literature [29,30,31,32].

Compounds	P_app_ (×10^−6^ cm/s)	K_p,uu,CSF_
Donepezil	41.6 ± 7.0	1.8 (human) ^1^
Midazolam	41.5 ± 2.8	1.35 (rat) ^2^
Carbamazepine	29.0 ± 4.4	0.979 (monkey) ^3^
Propranolol	24.6 ± 4.5	0.42 (human) ^3^
Phenytoin	21.8 ± 1.2	0.396 (rat) ^3^
Metoprolol	18.1 ± 0.28	0.93 (human) ^3^
Naproxen	12.3 ± 1.4	N/A
Sulpiride	10.2 ± 0.82	0.049 (rat) ^2^
Atenolol	7.6 ± 0.54	0.331 (monkey) ^3^
Dantrolene	7.1 ± 3.7	0.22 (monkey) ^3^
Fexofenadine	3.5 ± 0.32	0.0012 (mouse) ^4^
Rhodamine123	0.67 ± 0.14	N/A
FITC-dextran 4 kDa	0.70 ± 0.15	N/A
FITC-dextran 40 kDa	0.23 ± 0.062	N/A

K_p,uu,CSF_: unbound plasma-cerebrospinal fluid (CSF) partition coefficient. N/A: not available. ^1^ [29], ^2^ [30], ^3^ [31], ^4^ [32]. The data were calculated, wherein the provided ratios of the total concentration in plasma to cerebrospinal fluid and unbound fractions were multiplied to obtain the final values.

## Data Availability

Data are contained within the article and Appendix A.

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
