# Peer review of "An Improved In Vitro Blood-Brain Barrier Model for the Evaluation of Drug Permeability Using Transwell with Shear Stress"

_pharmaceutics, 2023, doi:10.3390/pharmaceutics16010048_

Round 1

Reviewer 1 Report

Comments and Suggestions for Authors

An Article “An Improved in vitro Blood-Brain Barrier Model for Evaluation of Drug Permeability using Transwell with Shear Stress” focuses on developing novel in vitro BBB model suitable for permeability assessment by applying shear stress to an high-throughput screening friendly transwell system using an orbital shaker. The research team concluded that this model provided robust conditions for evaluating the permeability of CNS drug candidates, potentially improving the reliability of in vitro BBB models in drug development. The work is relevant to current need

The comments are:

1.      Background of research work need to compare with available literature and model.

2.      Explain and compare how this model is novel and different from them along with hypothesis.

3.      2.6. Efflux transporter (P-gp, BCRP) inhibition assay. Cite this method, if not developed in house.

4.      Table 1 compounds. How the compounds selected/screed for permeability study?

5.      Line 365-366: When developing CNS drugs, it is crucial to evaluate whether candidate compounds can safely permeate the BBB. Is our aim to deliver safely, or evaluate the quantum of permeability?

Author Response

Thank you very much for taking the time to review this manuscript. Please find the detailed responses below and the corresponding revisions in track changes in the re-submitted files.

The comments are:

  1. Background of research work need to compare with available literature and model.

 → Thank you for your comment. We have added text in lines 51–53 and 407–413 to address this shortcoming.

  1. Explain and compare how this model is novel and different from them along with hypothesis.

 → Thank you for your comment. We have added text in lines 413–415 to address this shortcoming.

  1. 2.6. Efflux transporter (P-gp, BCRP) inhibition assay. Cite this method, if not developed in house.

 → Thank you for your comment. We have provided a reference to the method in line 166.

  1. Table 1 compounds. How the compounds selected/screed for permeability study?

→ Thank you for your comment. We initially compiled a list of drugs known to be well-delivered to the brain and those with poor brain penetration through literature search. Subsequently, we selected drugs with available LC-MS/MS methods for analysis.

  1. Line 365-366: When developing CNS drugs, it is crucial to evaluate whether candidate compounds can safely permeate the BBB. Is our aim to deliver safely, or evaluate the quantum of permeability?

 → Thank you for your comment. We have corrected the sentence in line 382 to clarify the aim of the study and the discussion.

Reviewer 2 Report

Comments and Suggestions for Authors

Dear Authors, I have two comments.

Comment 1;

In Fig. 2, 3 and 4, TEER values are shown, and seem to be much lower than those in vivo (3000~). The authors need to show the reasonability to use this in vitro model.

Comment 2;

Regarding the study on the BBB function with shear stress, describing other groups’ research activities will be helpful for readers to understand the significance of this manuscript. At least, Kurosawa et al. Pharm Res 39:1535-1547 (2022) was found in Pubmed.

Comments on the Quality of English Language

N/A

Author Response

Thank you very much for taking the time to review this manuscript. Please find the detailed responses below and the corresponding revisions in track changes in the re-submitted files.

Dear Authors, I have two comments.

Comment 1;

In Fig. 2, 3 and 4, TEER values are shown, and seem to be much lower than those in vivo (3000~). The authors need to show the reasonability to use this in vitro model.

→ Thank you for your comment. We have added text in lines 455–458 to address this shortcoming.

Comment 2;

Regarding the study on the BBB function with shear stress, describing other groups’ research activities will be helpful for readers to understand the significance of this manuscript. At least, Kurosawa et al. Pharm Res 39:1535-1547 (2022) was found in Pubmed.

→ Thank you for your comment. We have added text in lines 51–54 to address this shortcoming.

Reviewer 3 Report

Comments and Suggestions for Authors

Overall this article is a careful analysis of a novel culture technique for hCMEC/D3 cells to improve barrier integrity and potentially offer a means to perform HTS of molecules for drug discovery and development. Although it is unclear if this technique is adaptable for more than one plate at a time making it difficult to conceive of how HTS could be performed. 

The proposed system is a simple and well described accessible system. The cell lines used while are not the most cutting edge or potentially best model, the purpose of this work seems to be to establish a model that is adaptable for many labs. 

Minor concerns

What material of membrane was used in the transwell inserts? This was not clearly identified in the methods

Figure 2 describes a set of experiments to validate further studies, it’s not clear if panels c and d were performed in an 24 well plate or other container. It is also not clear if co culture was used or just monoculture

Similarly it is unclear through the rest of the experiments if co culture with the U87 cells was continued. 

It would be helpful to know if permeability and TEER measurements were taken in the AS device or if the inserts were inserted into a 24 well plate. 

The authors make a point to discuss the importance of the viscosity of the culture media, the authors should consider adding experiments using media containing starch or other additives to bring the viscosity closer to that of blood. 

Author Response

Thank you very much for taking the time to review this manuscript. Please find the detailed responses below and the corresponding revisions in track changes in the re-submitted files.

Overall this article is a careful analysis of a novel culture technique for hCMEC/D3 cells to improve barrier integrity and potentially offer a means to perform HTS of molecules for drug discovery and development. Although it is unclear if this technique is adaptable for more than one plate at a time making it difficult to conceive of how HTS could be performed.

→ Thank you for your comment. Our technique that uses an annular shaking dish allows for the simultaneous cultivation of multiple samples. We have added text to clarify this in lines 459–461.

The proposed system is a simple and well described accessible system. The cell lines used while are not the most cutting edge or potentially best model, the purpose of this work seems to be to establish a model that is adaptable for many labs.

Minor concerns

What material of membrane was used in the transwell inserts? This was not clearly identified in the methods

→ Thank you for your comment. We have added the materials used in lines 106, 224, and 244.

Figure 2 describes a set of experiments to validate further studies, it’s not clear if panels c and d were performed in an 24 well plate or other container. It is also not clear if co culture was used or just monoculture

→ Thank you for your comment. We have added text in lines 230 and 244 for clarity.

Similarly it is unclear through the rest of the experiments if co culture with the U87 cells was continued.

→ Thank you for your comment. We have added text in lines 273–275 for clarity.

It would be helpful to know if permeability and TEER measurements were taken in the AS device or if the inserts were inserted into a 24 well plate.

→ Thank you for your comment. We have added text in lines 275–276 for clarity.

The authors make a point to discuss the importance of the viscosity of the culture media, the authors should consider adding experiments using media containing starch or other additives to bring the viscosity closer to that of blood.

→ Thank you for your comment. We have added text in lines 452–454 to highlight this as a limitation of our study.
